# communications
# engineering

# Clustering method for time-series images using quantum-inspired digital annealer technology

Tomoki Inoue [1], Koyo Kubota [1], Tsubasa Ikami [2], Yasuhiro Egami [3], Hiroki Nagai [2], Takahiro Kashikawa[4], Koichi Kimura[4] & Yu Matsuda [1✉]

Time-series clustering is a powerful data mining technique for time-series data in the absence of prior knowledge of the clusters. Here we propose a time-series clustering method that leverages an annealing machine, which accurately solves combinatorial optimization problems. The proposed method facilitates an even classification of time-series data into closely located clusters while maintaining robustness against outliers. We compared the proposed method with an existing standard method for clustering an online distributed dataset and found that both methods yielded comparable results. Furthermore, the proposed method was applied to a flow measurement image dataset containing noticeable noise with a signal-to-noise ratio of approximately unity. Despite a small signal variation of approximately 2%, the proposed method effectively classified the data without any overlaps among the clusters. In contrast, the clustering results of the existing methods exhibited overlapping clusters. These results indicate the effectiveness of the proposed method.

[1] Department of Modern Mechanical Engineering, Waseda University, 3-4-1 Ookubo, Shinjuku-ku, Tokyo 169-8555, Japan. [2] Institute of Fluid Science, Tohoku University, 2-1-1 Katahira, Aoba-ku, Sendai, Miyagi-prefecture 980-8577, Japan. [3] Department of Mechanical Engineering, Aichi Institute of Technology, 1247 Yachigusa, Yakusa-Cho, Toyota, Aichi-prefecture 470-0392, Japan. [4] Quantum Application Core Project, Quantum Laboratory, Fujitsu Research, Fujistu Ltd, Kawasaki, Kanagawa 211-8588, Japan. ✉email: y.matsuda@waseda.jp

The collection of large-sized datasets has drastically increased with advancements in data storage and data acquisition technologies. Time-series data containing one or multiple variables (e.g., images) that vary with time is extensively recorded and analyzed in various fields, such as science, engineering, medical science, economics, and finance[1–3]. Clustering is a powerful data mining technique for classifying these data into related groups in the absence of sufficient prior knowledge of the groups[4–6]. In particular, when dealing with time-series data, the clustering technique is referred to as time-series clustering[7–9]. Many studies on time-series clustering have been summarized in review papers[2,7–11]. In addition, several libraries for time-series clustering have been made available on the web[12–16] and are widely used. Following the literature[7,8], time-series clustering is defined as "the process of unsupervised partitioning a given time-series dataset into clusters, in such a way that homogenous time-series data are grouped together based on a certain similarity measure, is called time-series clustering." Three main methods are commonly employed for time-series clustering: raw-data-based/shape-based, feature-based, and model-based approaches[7,8]. As an example, the raw-data-based/shape-based approach is illustrated in Fig. 1. These methods differ in their initial calculation procedures. The raw-data-based/shape-based approach directly uses the raw data for clustering, whereas the feature-based approach transforms the raw data into a low-dimensional feature vector. The model-based approach assumes that the time-series data are generated from a stochastic process model, and the parameters of the model are estimated from the data. The raw-data-based and feature-based approaches are more commonly used because the performance of the model-based approach degrades when clusters are close to each other[2,8]. The subsequent step involves calculating the similarity or distance between two data points, feature-vectors, or models. Then, the data is grouped into clusters based on the measured similarity or distance using machine learning methods. Clustering algorithms commonly employed for time-series data include partitioning, hierarchical, model-based, and density-based clustering algorithms[7,8]. Among partitioning clustering algorithms, k-means clustering is one of the most widely used algorithms[5,6,17,18]. Its

main advantage lies in its low computational cost. However, the method requires user to pre-determine a number of clusters. In a hierarchical clustering algorithm, the number of clusters does not need to be pre-determined. However, once clusters are split or merged using the divisive or agglomerative methods, they cannot be adjusted. Neural network approaches such as self-organizing maps[19] and hidden Markov model[5] are employed as model-based clustering approaches. In addition to the above-mentioned disadvantage of the performance degradation for close clusters, these approaches suffer from high computational costs. Density-based methods, such as density-based spatial clustering of applications with noise (DBSCAN)[20], do not require users to pre-determine a number of clusters and is robust to outliers. However, in density-based methods, an appropriate choice of parameters is difficult, and it is known that they suffer from the curse of dimensionality. Overall, each method has its advantages and disadvantages. A definitive method that can be used for all datasets does not exist, and an appropriate method should be employed depending on the purpose and dataset to be processed. Recently, continued attempts have been made to improve the performance of each method. As examples, studies published in the last three years are introduced as follows: the extension of dynamic time warping (DTW)[21,22], the measure based on quantile cross-spectral density[23], and the measure of two linear fuzzy information granule time-series[24] have been proposed to calculate the similarity or distance. A clustering method that focuses on the time-varying moment was proposed for financial time-series data[25]. A model-based approach based on the mixture of linear Gaussian state space model was proposed[26]. A notable research trend is approaches based on deep learning[27–29], which is different from the previously mentioned unsupervised methods. As a contrasting approach, a computationally efficient approach based on single-template matching was proposed[30]. However, to the best of the authors' knowledge, no method has been reported to concurrently achieve a high clustering performance (e.g., classification of data points that are close to each other, robustness to outliers, etc.) and low computational cost.

In this study, we propose a time-series clustering method that can achieve a higher clustering performance and lower

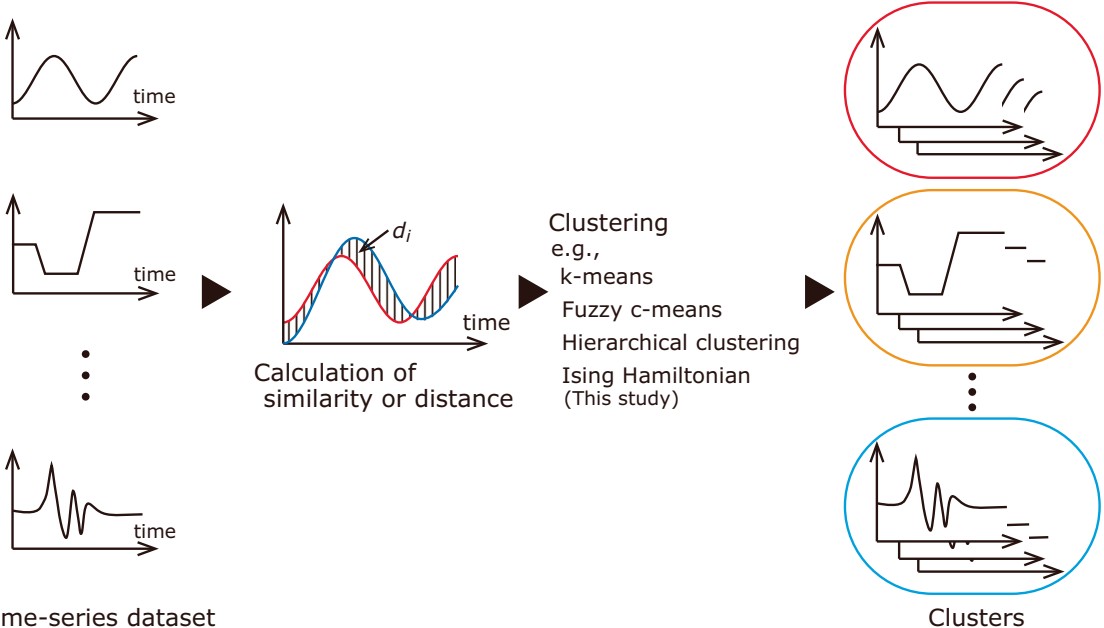

**Fig. 1 Typical clustering procedure of raw-data-based/shape-based approach.** The similarity or distance between two data points is calculated. The data points are classified into each cluster based on the similarity or distance.

computational cost. To achieve this goal, we focused on clustering algorithms using an annealing machine. As mentioned above, research has not included the study of clustering algorithms as much as the calculation of similarity or distance, with the exception of the use of deep learning. Annealing/Ising machines, such as quantum annealing and digital Ising machines, solve combinatorial optimization problems faster and more accurately than conventional computers[31–34]. Therefore, we expect that our proposed method can achieve clustering tasks that are challenging to achieve with existing methods. A unique characteristic of the proposed method, which is not found in existing methods, is its ability to evenly classify time-series data into closely related clusters while maintaining robustness against outliers. More specifically, the method can equally classify periodic time-series images into several phase ranges by assuming a sufficient number of images for each phase, given the long duration of the time-series data relative to the period. This paper provides a comprehensive explanation of our proposed method. We used the third-generation Fujitsu Digital Annealer (DA3), which is a quantum-inspired computing technology, for the clustering calculation. DA3 can solve quadratic unconstrained binary optimization (QUBO) problems, and the clustering problem can be formulated as an Ising model that is equivalent to a QUBO problem[35,36]. DA3 provides a solution in a large-scale problem space of up to 100 kbits. Subsequently, we applied our proposed method to two time-series datasets: one obtained from "the UEA & UCR time-series classification repository"[37–39], and the other consisted of flow measurement image data capturing the Kármán vortex street, periodic wakes, obtained in our previous data[40–42]. We specifically chose flow measurement data because it is typically high dimensional ($\sim 10^6$) and contains a measurement noise. For the clustering process, we employed raw-data-based and feature-based approaches. Furthermore, we compared our results with those obtained from existing standard methods, specifically "tslearn"[12] available online, and the conditional image sampling (CIS) method[43,44] (only for flow measurement data).

## Results and discussion

**Clustering of online available time-series dataset.** We demonstrated the application of the proposed method to classify the "crop" dataset available from the UEA & UCR time-series classification repository[37–39]. The clustering results obtained using the "TimeSeriesKMeans" function in "tslearn" and the proposed methods are shown in Fig. 2. The "crop" dataset contained 24 clusters. However, we present the results of two representative clusters. In this dataset, the correct classifications were known and displayed in Fig. 2. In addition, ensemble-averaged data for each method were calculated. As shown in Fig. 2a, the proposed method successfully classified the data, whereas the results obtained by the standard existing method (tslearn) exhibited some unfavorable classifications. We calculated the root mean squared error (RMSE) between the ensemble-averaged data of the correct data and those obtained by the proposed method and "tslearn". The RMSEs of the proposed and existing methods shown in Fig. 2a were 0.115 and 0.121, respectively. This further confirmed that the proposed method surpassed the standard existing method. On the other hand, the RMSEs of the proposed and the existing methods shown in Fig. 2b were 0.117 and 0.096, respectively. In this condition, the result obtained by the proposed method was inferior to that of the existing method. However, since the variance of the correct data is large, as shown in Fig. 2b, the classification is inherently difficult. The demonstrations for other datasets are provided in Supplementary Note 1. Consequently, we can conclude that the results of the proposed method are comparable to those of conventional methods.

**Clustering of flow measurement time-series dataset.** We applied our method to the flow measurement dataset of the Kármán vortex street to demonstrate its effectiveness for noisy data. A typical data of a snapshot is shown in Fig. 3a, and the image shows that the data contain noticeable noise with a signal-to-noise ratio (SNR) of approximately 1. The dimensions of the measurement area are shown in Fig. 3b. This periodic time-series dataset should be equally classified into each phase range because a sufficient number of images were acquired for each phase owing to the long duration of the measurement relative to the period. Therefore, this is a typical dataset to demonstrate the effectiveness of this method. In this study, we classified this time-series data into nine clusters using the proposed method, "tslearn," and the CIS method. The clustering results are shown in Fig. 4, where the data are presented on a two-dimensional scatter plot using multi-dimensional scaling (MDS). In the MDS calculation, the distance between the data $\mathbf{x}_i$ and $\mathbf{x}_j$ is represented as $|\sin(\theta_{i,j}/2)|$, where $|a|$ represents the absolute value of $a$, and $\theta_{i,j}$ corresponds to the angle between data vectors $\mathbf{x}_i$ and $\mathbf{x}_j$. Since the Kármán vortex street dataset used in the analysis is a periodical phenomenon with a maximum distance of unity, the data points were distributed along a circle with a radius of 1/2. As illustrated in Fig. 4a, the proposed method successfully classified the data points without overlaps. The data points were evenly classified into each cluster, and the cluster sizes were similar, which is a favorable result. The data points outside the circle with a radius of 1/2 were considered outliers, which is reasonable because these data points were considered disturbances deviating from periodic phenomena. However, the outliers were classified into one of the clusters in the standard existing method (Fig. 4b). This will be inappropriate when calculating the ensemble average of the data. The CIS method only classified the data points on the circle as shown in Fig. 4c. However, some clusters exhibited overlapping regions and did not form discrete clusters. Density-based methods, such as DBSCAN, are known as powerful clustering methods. However, the data points on the circle were classified into a single cluster in DBSCAN.

The ensemble-averaged pressure distributions are shown in Figs. 5–7. The proposed method (Fig. 5) and the CIS method (Fig. 7) effectively extract a periodic vortex generation despite a small pressure variation of approximately 2%. On the other hand, the pressure distribution obtained from the standard method failed to accurately extract the periodic motion. For example, the vortex located at the upper side suddenly disappeared from phase 2 to phase 3, and the vortex at the upper side reversed its flow direction from phase 5 to phase 6 (Fig. 6). This discrepancy can be attributed to the overlapping clusters observed in Fig. 4b. As the pressure decreases when the vortex comes, we compared the minimum pressure at the center of the vortex between the proposed and CIS methods. The ensemble-averaged pressure values were $p/p_{\mathrm{ref}} = 0.982 \pm 0.001$ and $p/p_{\mathrm{ref}} = 0.984 \pm 0.002$ for the proposed and CIS methods, respectively, where the error represents the standard deviation and $p_{\mathrm{ref}}$ denotes the atmospheric pressure. The pressure obtained by the CIS method was slightly higher than that of the proposed method, which aligned with the observations in Figs. 5 and 7. The difference indicates that the vortex was weakened in the CIS method because of the previously mentioned overlapping clusters, where data from different phases were also included in the ensemble averaging process. These findings provide further evidence that the proposed method is a powerful clustering approach for analyzing periodic phenomena.

## Conclusions

We propose a novel clustering method using an annealing machine. We added a term that adjusts the number of data

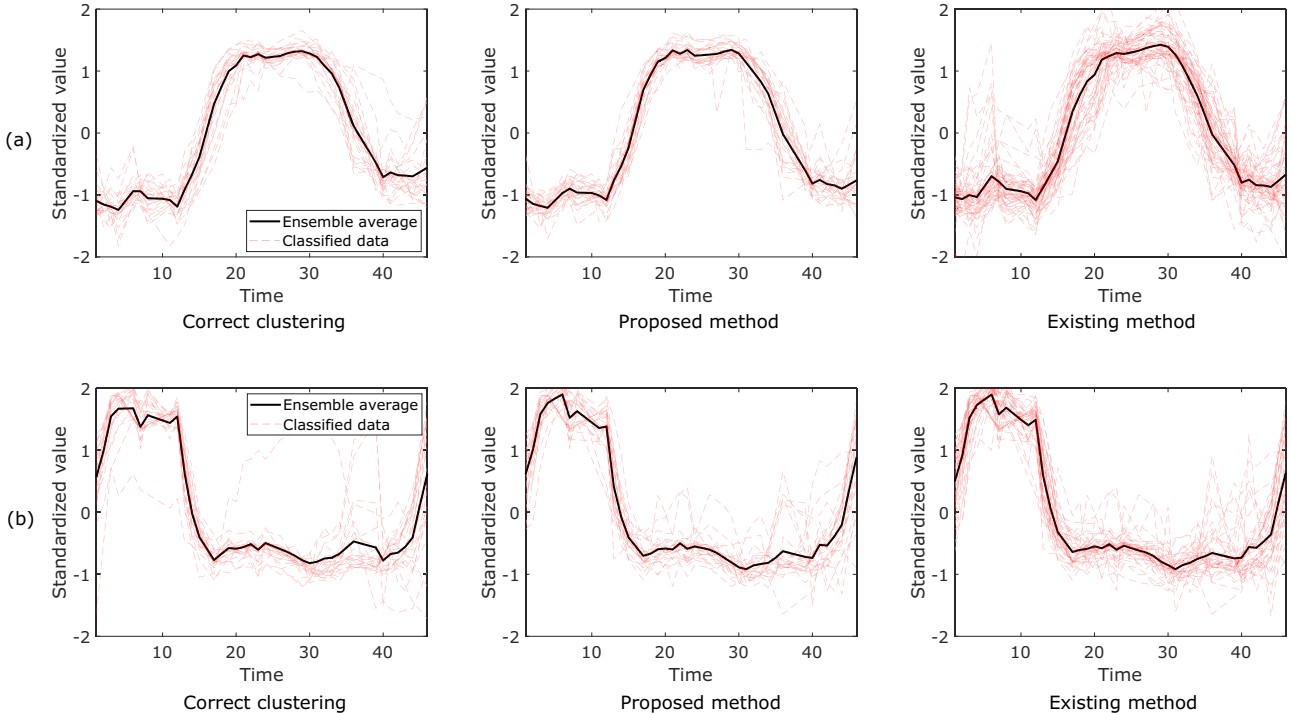

**Fig. 2 Typical clustering results for "crop" dataset from the UEA & UCR time-series classification repository using the proposed and existing methods.** The data labeled as class 1 and class 17 in the repository are shown in (**a**) and (**b**), respectively.

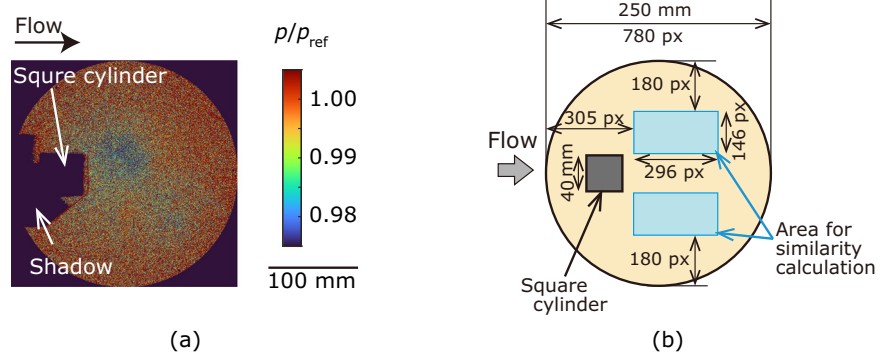

(a)  (b)

**Fig. 3 Typical raw data of PSP measurement and calculation condition.** (**a**) Typical pressure distribution, where pressure $p$ is normalized by an atmospheric pressure $p_{ref}$. Reproduced from Inoue et al.[42]. (**b**) Dimensions of experimental setup and area for similarity calculation.

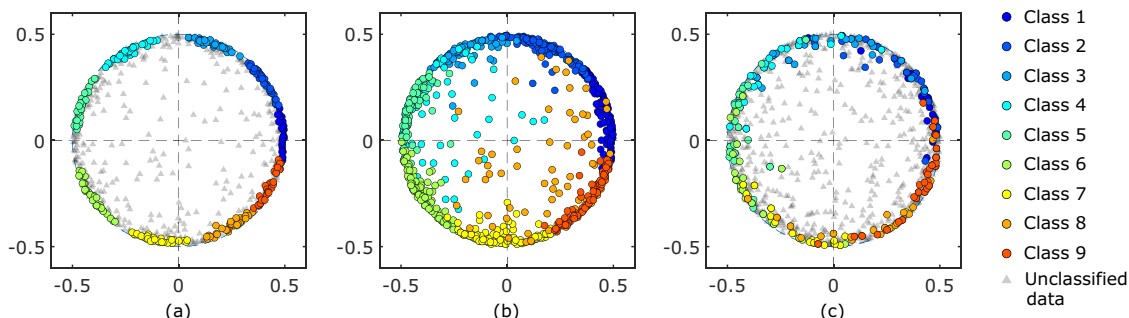

(a)  (b)  (c)

**Fig. 4 Clustering results shown in two-dimensional scatter plot based on MDS.** (**a**) The result by the proposed method, (**b**) that by the existing standard method (tslearn), (**c**) that by the conditional image sampling (CIS) method.

classified into each cluster to a QUBO model. In this study, we applied our proposed method to two distinct datasets: one is the "crop" dataset available from the UEA & UCR time-series classification repository and the other is a flow measurement image dataset obtained in our previous study. For the clustering of "crop" dataset, we also employed a standard existing method distributed as "tslearn," in which the distance between each data was calculated based on the Euclidean distance and the clustering

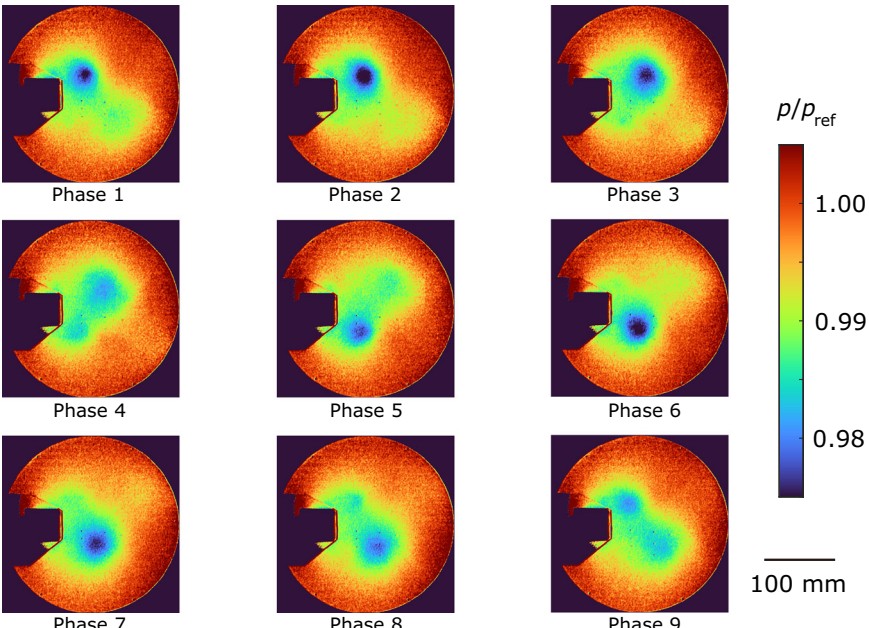

**Fig. 5 Ensemble-averaged pressure distribution for the proposed method.** The images are in phase order, and the vortices are flowing in this order. Pressure $p$ is normalized by an atmospheric pressure $p_{ref}$.

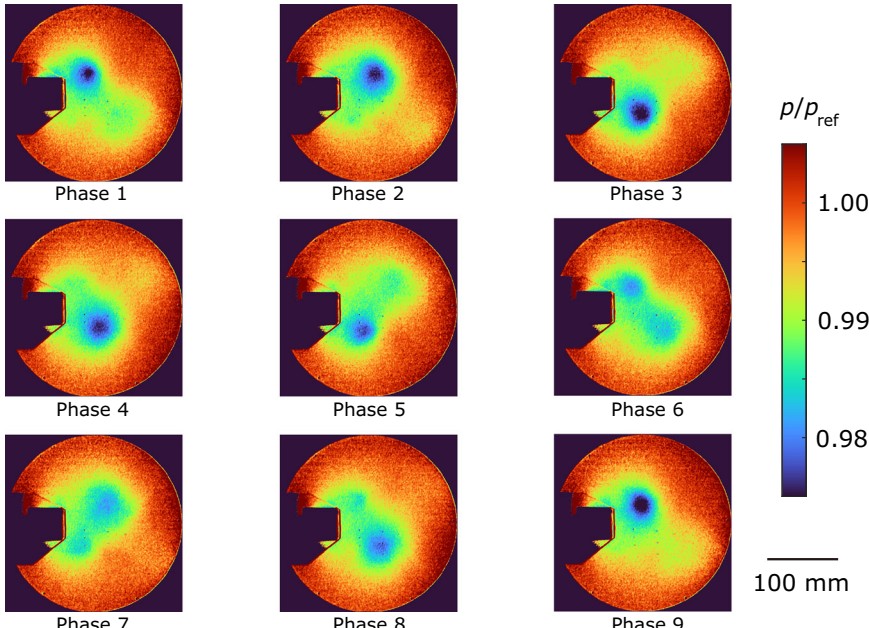

**Fig. 6 Ensemble-averaged pressure distribution for the existing standard method (tslearn).** The vortices are not flowing in the phase order. Pressure $p$ is normalized by an atmospheric pressure $p_{ref}$.

was calculated by the k-means++ algorithm. Comparing the results obtained from our proposed method and the existing method, we observed that the variation of the data points obtained by the proposed method was smaller than that by the existing method. In this dataset, the correct clustering result was provided. Then, we calculated the ensemble-averaged data, and the root mean squared errors (RMSEs) between the correct data and the ensemble-averaged data were compared. Our findings indicate that both methods provide similar results for this dataset.

Next, we applied our clustering method to the flow measurement image dataset, which consisted of the time-series pressure distributions induced by the Kármán vortex street. This dataset exhibited periodicity. Another characteristic of this data is that the dataset contains a noticeable noise with a signal-to-noise ratio of approximately 1. For comparison, the dataset was also classified using the standard existing method and the conditional image sampling (CIS) method, which is specifically designed for flow measurement data. The proposed method successfully classified the data without any overlap between the clusters in spite of the small pressure variation of approximately 2%. On the other hand, both the existing and the CIS methods exhibited overlapping of clusters, failing to form discrete clusters. In particular, the overlap between the clusters calculated by the existing method was large; thus, the vortex suddenly disappeared at times and

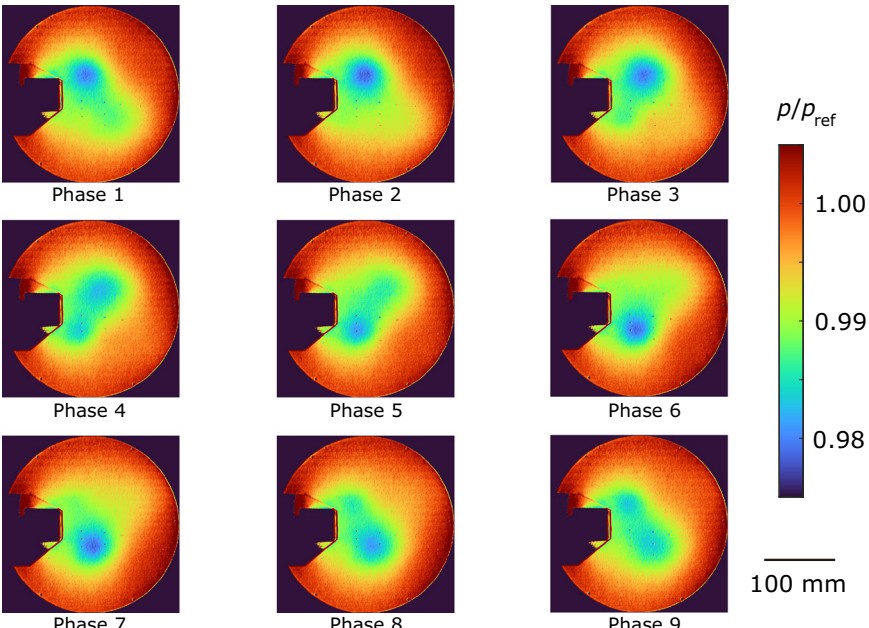

**Fig. 7 Ensemble-averaged pressure distribution for the conditional image sampling (CIS) method.** The vortices are weaker than those of the proposed method. Pressure $p$ is normalized by an atmospheric pressure $p_{ref}$.

exhibited reverse flow at other times in the ensemble-averaged pressure distribution. It was also found that the vortex was weakened in the ensemble-averaged pressure distribution obtained by the CIS method. These outcomes highlight the superior performance of the proposed method in the clustering periodic phenomena. The clustering algorithm using an annealing machine is a promising algorithm for large dataset. However, the calculation of similarity or distance is conducted by conventional computers. This is considered to be a major limitation that needs to be resolved when handling large datasets.

## Methods

**Proposed method for time-series clustering.** We propose a clustering method using an annealing machine. We focused on the raw-data-based and feature-based approaches for time-series data analysis. We considered a clustering problem that a given dataset of $n$ time-series data $\mathbf{X} = \{\mathbf{x}_1, \mathbf{x}_2, \cdots, \mathbf{x}_n\}$, where $\mathbf{x}_i$ is a column vector, is classified to $k$ clusters $c = \{c_1, c_2, \cdots, c_k\}$. Since DA is designed to solve QUBO problems, an objective function is expressed as a QUBO problem. The Hamiltonian for the clustering problem is described as follows[45,46]:

$$\mathcal{H} = \sum_c \sum_{i \neq j} d_{i,j} q_{g,i} q_{g,j} - \lambda_1 \sum_{\mathbf{X}} \left( \sum_c q_{g,j} - 1 \right)^2 \quad (1)$$

where $q_{g,i} = 1$ when $\mathbf{x}_i$ belongs to cluster $c_g$ and $q_{g,i} = 0$ when $\mathbf{x}_i$ does not belong to the cluster $c_g$, that is,

$$q_{g,i} = \begin{cases} 1 : \mathbf{x}_i \in c_g \\ 0 : \mathbf{x}_i \notin c_g \end{cases} \quad (2)$$

The similarity or inverse of the distance between $\mathbf{x}_i$ and $\mathbf{x}_j$ is denoted as $d_{i,j}$, and $\lambda_1$ is a hyperparameter. The sum $\sum_{i \neq j} d_{i,j} q_{g,i} q_{g,j}$ represents the sum of the similarity or the inverse of the distance between two data points belonging to a cluster. The sum $\sum_c$ represents the sum over all clusters in the first term

of Eq. (1). Clustering can be calculated by minimizing $-\mathcal{H}$, i.e.,

$$\min - \sum_c \sum_{i \neq j} d_{i,j} q_{g,i} q_{g,j} + \lambda_1 \sum_{\mathbf{X}} \left( \sum_c q_{g,j} - 1 \right)^2 \quad (3)$$

The second term in Eq. (3) represents a constrained term ensuring each data point belongs to only one cluster[45,46]. The value $\lambda_1$ determines the strictness of this constraint, where a smaller value enables some data points to be treated as outliers and not assigned them to any cluster. This study considered the following minimization problem:

$$\min - \sum_c \sum_{i \neq j} d_{i,j} q_{g,i} q_{g,j} + \lambda_1 \sum_{\mathbf{X}} \left( \sum_c q_{g,j} - 1 \right)^2 + \lambda_2 \sum_c \left( \sum_j q_{g,j} \right)^2 \quad (4)$$

where the third term in Eq. (4) adjusts the number of data points classified into each cluster. We denote $S_g = \sum_j q_{g,j}$ to simplify the notation, indicating the number of data points belonging to the cluster $c_g$. Then, the third term of Eq. (4) is written as

$$\sum_c \left( \sum_j q_{g,j} \right)^2 = \sum_c S_g^2 \quad (5)$$

The mean number of data points and the variance of data points belonging to each cluster are represented by $\mu$ and $\sigma^2$, respectively. Eq. (5) is written as

$$\sum_c S_g^2 = \sum_c \left\{ \left( S_g - \mu \right)^2 + 2 S_g \mu - \mu^2 \right\} = k \left( \sigma^2 + \mu^2 \right) \quad (6)$$

When $m$ data points are classified into one of $k$ clusters, the mean $\mu = m/k$ is a constant. Then, as the variance decreases, i.e., the third term in Eq. (4) becomes smaller, the data points are evenly classified into each cluster. As the number of data points classified into each cluster decreases, the mean $\mu$ decreases and the third term also becomes smaller. In other words, adding this term enables us to easily adjust the number of data points in each cluster by only varying $\lambda_2$. The effect of $\lambda_2$ on the clustering of the flow measurement dataset is discussed in Supplementary Note 2. This adjustment is difficult for many existing clustering algorithms.

**Time-series dataset for demonstration**. We applied the proposed clustering method to two time-series datasets. One of the datasets, named "crop," was obtained from the UEA & UCR time-series classification repository[37–39]. These time-series data were derived from images taken by the FORMOSAT-2 satellite. The dataset consists of 24 classes corresponding to an agricultural land-cover map, and each data point corresponds to its temporal evolution. The time-series length was 46, and the data were one-dimensional. The data were standardized to have a mean of 0 and a variance of 1. We compared the clustering results obtained by the proposed method and those obtained by "tslearn."[12] In this study, we used the "TimeSeriesKMeans" function in "tslearn." The parameters in the function were set to general settings as follows: the number of clusters was 24, the metric (distance between each data) was Euclidean, the method for initialization was k-means++, and the other parameters were employed default values. This is a standard time-series clustering method. In the proposed method, the Euclidean distance was also used as the metric, and the inverse of the metric was used to minimize the first term in Eq. (4). The data were multiplied by $10^4$ before being transferred to DA3 because it can only handle integer values. Since all data points should belong to one of the clusters in this dataset, the parameter $\lambda_1$ was approximately 100 times larger than $\lambda_2$. The actual values used for the calculation are shown in the code attached in Supplementary Note 3. In this condition, a solution that all data points belonged to one of the clusters (the second term of Eq. (4) was 0) was obtained.

The second dataset used in this study was the flow image data obtained in our previous study[40–42], which were measured using the pressure-sensitive paint (PSP) method[47–49]. The PSP method is a pressure distribution measurement technique based on the oxygen quenching of the phosphorescence emitted from the dyes incorporated into the PSP coating. The measured data were the pressure distribution induced by the Kármán vortex street behind a square cylinder as shown in Fig. 3a. The data size was $780 \times 780$ spatial grids. The flow velocity was 50 m/s, and the Reynolds number was $1.1 \times 10^5$. The number of data points was 720. The pressure difference was too small to be detected using the PSP technique because of the small variation in the phosphorescence intensity. Then, the measured pressure contained noticeable noise, and the noise should be reduced from the data. It is well known that the Kármán vortex is a periodic phenomenon. The data were classified into several phase groups and averaged within these groups to reduce the noise and extract useful patterns, which is one of the purposes of time-series clustering. The cosine similarity measure was used to assess the similarity between the data because we focused on the phase information of the vortex. Since the PSP data were a time-series image data with two spatial dimensions and one temporal dimension, the pressure distribution data were reshaped into a column vector. Consequently, the time-series PSP data are written as $n$ time-series data $\mathbf{X} = \{\mathbf{x}_1, \mathbf{x}_2, \cdots, \mathbf{x}_n\}$, where $\mathbf{x}_i$ is a vector corresponding to a reshaped pressure distribution. Since the measured PSP data contains significant noise of SNR ~ 1, the denoised data was used for the calculation of the similarity. Following the literature[50], the dataset with small noise can be obtained by considering the truncated singular value decomposition (SVD). We considered a data matrix $\mathbf{Y} = [\mathbf{x}_1 \, \mathbf{x}_2 \cdots \mathbf{x}_n]$, where the data matrix $\mathbf{Y}$ was obtained by arranging vectors $\mathbf{x}_i$ in time-series order. SVD provides the following representation:

$$\mathbf{Y} = \mathbf{U}\boldsymbol{\Sigma}\mathbf{V} \tag{7}$$

where the matrices $\mathbf{U}$ and $\mathbf{V}$ are unitary matrices, and the superscript $\mathsf{T}$ shows the transpose. The matrix $\boldsymbol{\Sigma}$ is a diagonal matrix of singular values arranged in descending order. It is well known that the data can be approximated by a truncated SVD[51]

as follows:

$$\widetilde{\mathbf{Y}} = \widetilde{\mathbf{U}}\widetilde{\boldsymbol{\Sigma}}\widetilde{\mathbf{V}} \tag{8}$$

where $\widetilde{\boldsymbol{\Sigma}}$ is a first $r \times r$ diagonal matrix and $r$ is a truncation rank. The matrices $\widetilde{\mathbf{U}}$ and $\widetilde{\mathbf{V}}$ are reduced matrices corresponding to $\widetilde{\boldsymbol{\Sigma}}$. Then, we obtained the noise-reduced time-series data matrix of $\widetilde{\mathbf{Y}} = [\widetilde{\mathbf{x}}_1 \, \widetilde{\mathbf{x}}_2 \cdots \widetilde{\mathbf{x}}_n]$ or the time-series data of $\widetilde{\mathbf{X}} = \{\widetilde{\mathbf{x}}_1, \widetilde{\mathbf{x}}_2, \cdots, \widetilde{\mathbf{x}}_n\}$. We set $r = 5$, which is a commonly used truncation value. Subsequently, the cosine similarity $\cos \theta_{i,j}$ was calculated as follows:

$$\cos \theta_{i,j} = \frac{\langle \widetilde{\mathbf{x}}_i, \widetilde{\mathbf{x}}_j \rangle}{||\widetilde{\mathbf{x}}_i||_2 ||\widetilde{\mathbf{x}}_j||_2} \tag{9}$$

where $\langle \widetilde{\mathbf{x}}_i, \widetilde{\mathbf{x}}_j \rangle$ is the inner product of $\widetilde{\mathbf{x}}_i$ and $\widetilde{\mathbf{x}}_j$, and $||\widetilde{\mathbf{x}}_i||_2$ is the $\ell 2$ norm of $\widetilde{\mathbf{x}}_i$. In the similarity calculation, we only considered the pressure distribution behind the square cylinder to reduce the computational cost (see Fig. 3b). Substituting $d_{i,j} = \cos \theta_{i,j}$ in Eq. (4), we calculated the clustering using DA3. Since the data were also multiplied by $10^4$ before being transferred to DA3, $d_{i,j} \sim 10^3$. The parameter $\lambda_1$ was 40 times larger than $\lambda_2$ ($\lambda_1 = 1 \times 10^6$), ensuring that each term in Eq. (4) was of a similar magnitude. In this condition, some data were classified as outliers. The images within the same cluster were ensemble averaged to extract useful patterns. Here, we note that the original image data $\mathbf{X}$ was averaged to extract the patterns, while the truncated dataset of $\widetilde{\mathbf{X}}$ was not used.

Considering that $\cos \theta_{i,j}$ lies within the range of $-1$ to 1, we introduced the following relation $r_{i,j}$, which range from 0 to 1:

$$
\begin{aligned}
r_{i,j} &= \frac{\cos \theta_{i,j} + 1}{2} \\
&= \cos^2 \frac{\theta_{i,j}}{2} \\
&= 1 - \sin^2 \frac{\theta_{i,j}}{2}
\end{aligned}
\tag{10}
$$

where $1 - r_{i,j} = 0$ when $i = j$ (the same data). Then, we define the distance metric between the data as $|\sin(\theta_{i,j}/2)|$, where $|a|$ represents the absolute value of $a$ and $\theta_{i,j}$ is the angle between data vectors. The maximum value of the distance is unity in this distance metric. This distance metric was used in the MDS calculation.

The time-series data were also classified by the "TimeSeriesKMeans" function in "tslearn" described above. In addition, we used the CIS method[43,44], which is a specialized methods designed specifically for PSP measurements. In the CIS method, the time-series data were classified into several phase groups based on the pressure data measured by a pressure transducer sensor which is a point sensor with a higher sampling rate than PSP. In other words, the CIS method requires an additional sensor for clustering. This reliance on an extra sensor can be considered one of the limitations of the CIS method.

## Data availability

The dataset, named "crop," was obtained from the UEA & UCR time-series classification repository http://timeseriesclassification.com/description.php?Dataset=Crop. The flow measurement dataset is available in Zenodo with identifier https://doi.org/10.5281/zenodo.10215642.

## Code availability

The code for the time-series clustering developed in this study is included in Supplementary Note 3. The code for computing the observation matrix from time-series data is available in Zenodo with identifier https://doi.org/10.5281/zenodo.10215642.

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

## Acknowledgements

The authors would like to express their gratitude to Dr Yasuhumi Konishi, Hiroyuki Okuizumi and Yuya Yamazaki for their assistance during the wind tunnel testing conducted at the Institute of Fluid Science, Tohoku University. The authors would also like to acknowledge Takafumi Oyama for the insightful discussions. We also gratefully appreciate Tayca Corporation for providing the titanium dioxide. Finally, we would like to thank Editage (www.editage.com) for English language editing. Part of the work was conducted under the Collaborative Research Project J23I041 of the Institute of Fluid Science, Tohoku University.

## Author contributions

T. Inoue: conceptualization, data curation, formal analysis, investigation, methodology, software, validation, visualization. K. Kubota: formal analysis, investigation, methodology, software, validation, visualization. T. Ikami: data curation, investigation, resources, visualization, writing—review and editing. Y.E.: data curation, formal analysis, investigation, resources, writing—review and editing. H.N.: investigation, resources, writing—review and editing. T.K.: methodology, software, validation. K. Kimura: methodology, software, validation. Y.M.: conceptualization, data curation, formal analysis, funding acquisition, investigation, methodology, project administration, resources, software, supervision, validation, visualization, writing—original draft, writing—review and editing.

## Competing interests

The authors declare the following competing interests: T.K. and K. Kimura are employees of Fujitsu Ltd. All other authors declare no competing interests.
