## [Peer Review File · Communications Engineering]

Reviewers' comments:

Reviewer #1 (Remarks to the Author):

I find this manuscript entitled: Clustering Method for Time-Series Images Using Quantum Inspired Computing Technology, to have considerable potential for publication as a valuable paper. Nevertheless, there are several factors that, if addressed, could further enhance the quality and impact of this work. The following outlines these factors:

- The paper would benefit from clearly defining its research objectives and outlining the specific contributions that it aims to make. This will provide readers with a clear understanding of the paper's scope and focus.
- I would highly recommend including a system diagram to help readers understand the study clearly.
- A more in-depth literature review could strengthen the paper's background and contextualize the research within the existing body of knowledge. Ensuring that relevant prior work is adequately covered will highlight the novelty and significance of the proposed research.
- Incorporating citations to recently published articles will undoubtedly enhance the paper's relevance within the context of existing related works. By referencing up-to-date and cutting-edge research, the paper will showcase its connection to the most current developments and findings in the field.
- Highlighting all possible potential limitations of the current work will add credibility to the research. Openly discussing constraints and possible sources of bias will enhance the paper's transparency too.
- The replication of current experiments and studies holds paramount importance in the scientific research community. So to ensure the credibility and reliability of research findings, explicit documentation of all experimental settings and parameters is imperative. Authors should endeavour to provide comprehensive details of the experimental setup, methodology, and relevant variables, that would enable subsequent validation of the presented results by their peers.

Reviewer #2 (Remarks to the Author):

This paper proposes a quantum-inspired clustering method using a Fujitsu Digital Annealer for time series. Although some interesting results are obtained from several real-world time series datasets, I believe this paper needs significant improvement to reach the acceptance bar.

1. The motivation is not clear. The paper only enumerates existing time series clustering methods, but fail to explain their relationship with the proposed method. Like, what problems existing methods have and how does the proposed method address them? The paper mentions manual parameter searching and high-dimensionality, but they are open challenges and have been widely discussed in the literature. Moreover, it is not clear how the proposed method solves such challenges.

2. The paper seems to emphasis "quantum-inspired" as the novel part of the proposed method; however, I can hardly capture such novelty due to the inappropriate organization of this paper. For example, "Hamiltonian" seems a very important concept of the proposed method, but this paper does not discuss its physical meaning, how it can help with time series clustering, and etc. That makes it difficult to understand why the method is "quantum-inspired".

3. The method discussed in "Proposed method for time-series clustering" section seems a general clustering method, and does not consider specific characteristic of time series.

Response to Reviewer #1

We thank the reviewer for carefully reading our manuscript and providing positive comments. We have answered your comment in the following:

1. The paper would benefit from clearly defining its research objectives and outlining the specific contributions that it aims to make. This will provide readers with a clear understanding of the paper's scope and focus.

Response:

We appreciate the reviewer's important suggestion. As per the comments, we revised the second paragraph (lines 12 to 15 on page 3) of the "Introduction" to better clarify the scope and contribution of the proposed methodology.

The scope of this paper is "the proposal of a time-series clustering method that can achieve a higher clustering performance and lower computational cost." To achieve the goal, we focused on clustering algorithms using an annealing machine. A unique characteristic of the proposed method is its ability to evenly classify time-series data into closely related clusters while maintaining robustness against outliers. The adjustment of the number of data points in each cluster is difficult in other clustering methods.

2. I would highly recommend including a system diagram to help readers understand the study clearly.

Response:

We are grateful for the reviewer's recommendation to improve our manuscript. We added a new figure (Fig. 1) to help readers understand the procedure of the clustering calculation.

3. A more in-depth literature review could strengthen the paper's background and contextualize the research within the existing body of knowledge. Ensuring that relevant prior work is adequately covered will highlight the novelty and significance of the proposed research.

Response:

We thank the reviewer for providing the valuable suggestion. In addition to the response to the first comment, we cited additional literature to emphasize the novelty and significance of this study in lines 16 to 21 on page 3.

4. Incorporating citations to recently published articles will undoubtedly enhance the paper's relevance within the context of existing related works. By referencing up-to-date and cutting-edge research, the paper will showcase its connection to the most current developments and findings in the field.

Response:

Following the reviewer's suggestion, in the "Introduction," we incorporated 11 recent studies (Refs. 11 and 21 to 30) to highlight the scope of the paper in lines 48 to 50 on page 2 and lines 1 to 10 on page 3, in addition to the responses to the first and third comments.

5. Highlighting all possible potential limitations of the current work will add credibility to the research. Openly discussing constraints and possible sources of bias will enhance the paper's transparency too.

Response:

In this study, the calculation of similarity or distance was conducted by conventional computers. This is considered to be a major limitation that needs to be resolved when dealing with large data sets. The sentence was added under the "Conclusion."

6. The replication of current experiments and studies holds paramount importance in the scientific research community. So to ensure the credibility and reliability of research findings, explicit documentation of all experimental settings and parameters is imperative. Authors should endeavour to provide comprehensive details of the experimental setup, methodology, and relevant variables, that would enable subsequent validation of the presented results by their peers.

Response:

According to the reviewer's suggestion, the code developed in this study is included in "Supplementary." Other conditions for the calculations have been added under "Materials and Methods." (e.g., Fig. 3(b))

Response to Reviewer #2

We appreciate the reviewer for carefully reading our manuscript and providing insightful comments. We have answered your comments as follows:

1. The motivation is not clear. The paper only enumerates existing time series clustering methods, but fail to explain their relationship with the proposed method. Like, what problems existing methods have and how does the proposed method address them? The paper mentions manual parameter searching and high-dimensionality, but they are open challenges and have been widely discussed in the literature. Moreover, it is not clear how the proposed method solves such challenges.

Response:

We thank the reviewer for providing the important suggestion. We cited and discussed recent literature (Refs. 11 and 21 to 30) in lines 48 to 50 on page 2 and lines 1 to 10 on page 3 in the “Introduction” to emphasize the novelty and significance of this study. We also revised lines 12 to 15 on page 3 in the “Introduction” to clarify the scope and contribution of the proposed methodology.

2. The paper seems to emphasize “quantum-inspired” as the novel part of the proposed method; however, I can hardly capture such novelty due to the inappropriate organization of this paper. For example, “Hamiltonian” seems a very important concept of the proposed method, but this paper does not discuss its physical meaning, how it can help with time series clustering, and etc. That makes it difficult to understand why the method is “quantum-inspired”.

Response:

We thank the reviewer for the comments. The original text was confusing; thus, we revised the text in lines 25 to 26 on page 3 and highlighted that “DA3 can solve quadratic unconstrained binary optimization (QUBO) problems, and the clustering problem can be formulated as an Ising model which is equivalent to a QUBO problem.” Moreover, we added the explanation in lines 1 to 2 on page 5 as “Since DA is designed to solve QUBO problems, an objective function is expressed as a QUBO problem.”

In this study, we focused on the usage of an annealing machine, which is a novel computer based on quantum-inspired technology. In the calculation by an annealing machine, an object function has to be written as a QUBO problem, which is practically equivalent to an Ising model in practice. We hope the revisions make the inspiration clear.

3. The method discussed in “Proposed method for time-series clustering” section seems a general clustering method, and does not consider specific characteristic of time series.

Response:

As the reviewer pointed out, the proposed method can be applied to time-series clustering as well as to other clustering problems. However, a unique characteristic of the proposed method that we want to emphasize is its ability to evenly classify time-series data into closely related clusters while maintaining robustness against outliers. In other words, the method can equally classify periodic time-series images into several phase ranges by assuming a sufficient number of images for each phase, particularly given the long duration of the time-series data relative to the period. Since the explanation was insufficient, additional explanations have been added to lines 18 to 23 on page 3 and lines 28 to 29 on page 5, respectively.

Reviewers' comments:

Reviewer #1 (Remarks to the Author):

The authors have taken into consideration my previous remarks, and I believe the paper is now suitable for acceptance in its present state.

Reviewer #2 (Remarks to the Author):

1. Since the proposed method is mostly based on quantum annealing machines, it is suggested to revise the title of the paper according, e.g., "...Using Quantum Annealing Machines".

2. UCR repository includes more than 80 datasets, only using one dataset may look cherry-picking. It is suggested to compare the proposed method with counterpart methods on more datasets.

3. It is suggested to provide a parameter analysis on the effect of λ^2 (weight of the balancing term) in Eq. (4), which is expected to show how the result varies with different λ^2 .

Response to Reviewer #1

We thank the reviewer for carefully reading and considering our manuscript for publication.

Response to Reviewer #2

We appreciate the reviewer for carefully reviewing our manuscript and providing insightful comments. We have answered your comments as follows:

1. Since the proposed method is mostly based on quantum annealing machines, it is suggested to revise the title of the paper according, e.g., "...Using Quantum Annealing Machines".

Response:

We thank the reviewer for the comments. In this study, we used Fujitsu digital annealer, which provides an alternative to quantum computing technology using a digital circuit design inspired by quantum phenomena. In other words, we did not use a quantum annealing machine itself, because the problem space that can be solved by a quantum machine is limited at present. Therefore, we believe that the term "quantum-inspired computing technology" is an accurate description.

2. UCR repository includes more than 80 datasets, only using one dataset may look cherry-picking. It is suggested to compare the proposed method with counterpart methods on more datasets.

Response:

We sincerely acknowledge the reviewer for the comment. We added examples of the application to two datasets in the Supplementary Note 1. We concluded that the results of the proposed method are comparable to those obtained using conventional methods.

3. It is suggested to provide a parameter analysis on the effect of λ_2 (weight of the balancing term) in Eq. (4), which is expected to show how the result varies with different λ_2 .

Response:

Indeed, as the reviewer pointed out, the choice of λ_2 affects the clustering result. Thus, we revised the explanation of λ_2 after Eq. (6) and added the clustering results in the Supplementary Note 2. As λ_2 increases, the number of data classified into clusters becomes equal and smaller. Once again, we appreciate the diligence of the reviewer.

REVIEWERS' COMMENTS:

Reviewer #2 (Remarks to the Author):

My concern has been addressed.

CMMSENG-23-0285B

Comments from Reviewer

Reviewer #2 (Remarks to the Author):

My concern has been addressed.

Response to Reviewer #2

We thank the reviewer for carefully reading and considering our manuscript for publication.